# POST-MORTEM ON A DEEP LEARNING CONTEST: A SIMPSON'S PARADOX AND THE COMPLEMENTARY ROLES OF SCALE METRICS VERSUS SHAPE METRICS

## ABSTRACT

To understand better good generalization performance in state-of-the-art neural network (NN) models, and in particular the success of the `AlphaHat` metric based on Heavy-Tailed Self-Regularization (HT-SR) theory, we analyze of a corpus of models that was made publicly-available for a contest to predict the generalization accuracy of NNs. These models include a wide range of qualities and were trained with a range of architectures and regularization hyperparameters. We break `AlphaHat` into its two subcomponent metrics: a scale-based metric; and a shape-based metric. We identify what amounts to a Simpson's paradox: where "scale" metrics (from traditional statistical learning theory) perform well in aggregate, but can perform poorly on subpartitions of the data of a given depth, when regularization hyperparameters are varied; and where "shape" metrics (from HT-SR theory) perform well on each subpartition of the data, when hyperparameters are varied for models of a given depth, but can perform poorly overall when models with varying depths are aggregated. Our results highlight the subtlety of comparing models when both architectures and hyperparameters are varied; the complementary role of implicit scale versus implicit shape parameters in understanding NN model quality; and the need to go beyond one-size-fits-all metrics based on upper bounds from generalization theory to describe the performance of NN models. Our results also clarify further why the `AlphaHat` metric from HT-SR theory works so well at predicting generalization across a broad range of CV and NLP models.

## 1 INTRODUCTION

It is of increasing interest to develop metrics to measure and monitor the quality of Deep Neural Network (DNN) models, especially in production environments, where data pipelines can unexpectedly fail, training data can become corrupted, and errors can be difficult to detect. There are few good methods which can readily diagnose problems at a layer-by-layer level and in an automated way.

Motivated by this, recent work introduced the `AlphaHat` metric, (i.e., $\hat{\alpha}$), showing that it can predict trends in the quality, or generalization capacity, of state-of-the-art (SOTA) DNN models *without access to any training or testing data* (Martin et al., 2021)—outperforming other metrics from statistical learning theory (SLT) in a large meta-analysis of hundreds of SOTA models from computer vision (CV) and natural language processing (NLP). The $\hat{\alpha}$ metric is based on the recently-developed Heavy-Tailed Self-Regularization (HT-SR) theory (Martin & Mahoney, 2021; 2019; 2020), which is based on statistical mechanics and Heavy-Tailed (HT) random matrix theory. Further, being a weighted average of layer metrics, understanding why `AlphaHat` works will help practitioners to diagnose potential problems layer-by-layer.

In this paper, we evaluate the `AlphaHat` ($\hat{\alpha}$) metric (and its subcomponents) on a series of pre-trained DNN models from a recent contest ("the Contest") to predict generalization in deep learning (Jiang et al., 2020a;b). The Contest was interested in metrics that were "causally informative of generalization," and it wanted participants to propose a "robust and general complexity measure" (Jiang et al., 2020a;b). These Contest models were smaller and more narrow than those analyzed in the large-scale meta-analysis (Martin et al., 2021). However, for that narrower class of models, the Contest data was more detailed. There were models with a wider range of test accuracies, including models that generalize well, generalize poorly, and even models which appear to be overtrained. The models are partitioned into sub-groups of fixed depth, where regularization hyperparameters (and width) are varied. This more fine-grained set of pre-trained models lets us evaluate

the $\hat{\alpha}$ metric, and its subcomponents, across the opposing dimensions of depth and hyperparameter changes, and more finely than analyzed previously on SOTA models.

Our analysis here provides new insights—on how theories for generalization perform on well-trained versus poorly-trained models; on how this depends in subtle ways on what can be interpreted as implicit scale versus implicit shape parameters of the models learned by these DNNs; and on how model quality metrics depend on architectural parameters versus solver parameters. Most importantly, this work helps clarify why the `AlphaHat` metric performs so well across so many models.

**Background: Heavy-Tailed Self-Regularization (HT-SR) Theory.** HT-SR theory is a phenomenology, based on Random Matrix Theory (RMT), and motivated by the statistical mechanics of learning, that explains empirical results on the spectral (eigenvalue) properties of SOTA DNNs (Martin & Mahoney, 2021). (A detailed discussion of HT-ST theory can be found in Martin & Mahoney (2021); here we can only summarize the basics.) Empirical results (Martin & Mahoney, 2021; 2019; 2020) show that, for nearly all well-trained DNN models (in CV and NLP), the layer *Correlation Matrices* $\mathbf{X} = \frac{1}{N}\mathbf{W}^T\mathbf{W}$ are HT, in the sense of being well-fit to a Power Law (PL) or truncated PL distribution (even though individual matrices $\mathbf{W}$ are *not* HT elementwise). Moreover, HT-SR theory indicates that as training proceeds and/or regularization is increased, the HTness of the correlations (however it is measured) generally increases (Martin & Mahoney, 2021). Using these facts, HT-SR theory allows one to construct various generalization capacity metrics for DNNs (which are implemented in the `WeightWatcher` open source tool (wei, 2018)) that measure the model average HTness (`Alpha`, `AlphaHat`, `LogSpectralNorm`, etc.) as a proxy for generalization capacity.

For moderately and very HT weight matrices, it is possible to quantify the HTness using a standard PL fit of the ESD. In this case, smaller PL exponents ($\alpha$) correspond to heavier tails. Given a DNN weight matrix $\mathbf{W}$, $(N \times M, \ N \geq M)$, let $\lambda$ be an eigenvalue of the correlation matrix $\mathbf{X} = \frac{1}{N}\mathbf{W}^T\mathbf{W}$. The Empirical Spectral Density (ESD), $\rho(\lambda)$, is the just an empirical fit of the histogram of the $M$ eigenvalues. In looking at hundreds of models and thousands of weight matrices, the *tail* of the ESDs of well trained DNNs can nearly always be well fit to a PL distribution:

$$\rho_{tail}(\lambda) \sim \lambda^{-\alpha}, \ \ x_{min} \leq \lambda \leq x_{max}. \tag{1}$$

Here, $x_{max} = \lambda_{max}$ is the maximum eigenvalue of the ESD, and $x_{min}$ is the start, fit using the procedure of Clauset et al. (2009). The best fit is given by the KS-distance $D_{KS}$ (denoted `QualityOfAlphaFit` below). For models that generalize well, the fitted $\alpha \gtrsim 2.0$ for (nearly) every layer (Martin & Mahoney, 2021; 2019; 2020). Models that generalize better generally have a smaller (weighted) average $\alpha$, `AlphaHat` ($\hat{\alpha}$), when compared within an architecture series (VGG, ResNet, ...) or with different size data sets (GPT vs GPT2) (Martin et al., 2021).

Still one may ask, *"Why are the layer correlations matrices $\mathbf{X}$ HT, but the weight matrices $\mathbf{W}$ themselves are not ?"* For the ESD of $\mathbf{W}$ to be HT, either $\mathbf{W}$ must be HT elementwise, having many spuriously large elements $W_{i,j}$, and/or $\mathbf{X}$ simply has many large eigenvalues. It is well known that the well-regularized models should not contain spuriously large elements $W_{ij}$, and methods like $L2$-regularization attempt to ensure this. In contrast, the largest eigenvalues of $\mathbf{X}$ will correspond to eigenvectors with the most important non-random information. Consequently, $\lambda_{max}$ will become larger, not smaller, as more information is learned. (This is well known from RMT. However, it is also known in machine learning, as it is the basis for methods like Latent Semantic Analysis (LSA).) The HT-SR theory exploits these facts to build a theory of generalization for DNNs.

Using layer PL fits, one can define the HT-based `AlphaHat` metric for an entire DNN model. This metric can predict trends in the quality of SOTA DNNs, even without access to training or testing data (Martin et al., 2021). `AlphaHat` ($\hat{\alpha}$) is a weighted average over $L$ layers of two complementary metrics, the PL exponent $\alpha$ and the logarithm maximum eigenvalue $\log \lambda^{max}$:

$$\hat{\alpha} = \sum_{l=1}^{L} \alpha_l \log \lambda_l^{max} \tag{2}$$

`AlphaHat` is both a weighted-average `Alpha`, weighted by the *Scale* of the layer ESD ($\log \lambda_l^{max}$), and a weighted-average `LogSpectralNorm`, weighted by the *Shape* of the layer ESD($\alpha$). We therefore evaluate how these two subcomponents, the `Alpha` and `LogSpectralNorm` metrics, individually perform when varying the opposing dimensions of depth (number of layers $L$) and regularization hyperparameters ($\theta$), such as dropout, momentum, weight decay, etc.

In investigating "*why* `AlphaHat` works," we discovered that the `Alpha` and `LogSpectralNorm` metrics frequently display a *Simpson's paradox*. Being aware of challenges with designing good

contests and with extracting causality from correlation (Pearl, 2009), we did not use the causal metric provided by the Contest. Instead, we adopted a different approach: we identified Simpson's paradoxes within the Contest data; and we used this to understand better the `AlphaHat` metric from HT-SR theory. A *Simpson's paradox* can arise when looking for trends in a large data set containing of multiple sub-groups; it occurs when a trend for the total data set reverses–or disappears–when considering the sub-groups individually (Simpson, 1951; Bickel et al., 1975; Robinson, 2009; Kievit et al., 2013). Simpson's paradoxes are particularly relevant when one wants to understand the data and attribute *causal* interpretations to the underlying data or model.

Because `AlphaHat` combines two complementary metrics, the `LogSpectralNorm` and `Alpha`, we consider two classes of metrics that have been used to predict DNN model quality: norm-based metrics from SLT (`LogSpectralNorm`); and HT-based metrics from HT-SR theory (`Alpha`). The norm-based metrics describe the *Scale* associated with the model implicitly-learned by the training process, and they have been used to provide generalization theory upper bound on simple models. Empirically, they can perform in strange and counter-intuitive ways on even moderately-large realistic models (Jiang et al., 2019). The HT (or other PL) metrics describe the HTness of the layer correlations using the *Shape* of the implicitly-learned model, are related to HT-SR theory (Martin & Mahoney, 2021; 2019; 2020). The *Shape*-based `Alpha` metric has been conjectured to predict test accuracy when only varying regularization hyperparameters, and the `AlphaHat` metric was developed to improve `Alpha` by correcting for the *Scale* of different layers when comparing similar models of differing depth (i.e., VGG11 – VGG19) (Martin & Mahoney, 2021; Martin et al., 2021).

**Our main results.**    We apply these SLT and HT-SR metrics to predict generalization accuracies of the pre-trained models provided in the Contest, as a function of both model depth $L$ and regularization hyperparameters $\theta$. Our main contributions are the following.

**New shape-based metrics.** Based on our preliminary analysis of the data, we introduce two new data-free quality metrics—`Alpha` and `QualityOfAlphaFit`, motivated by the HT-SR theory.

**Existence of Simpson's paradox in Contest data.** We examine models provided by the Contest, and we identify Simpson's paradoxes. Depending on the specific Contest task and model sub-group, `LogSpectralNorm` and `Alpha` are either strongly anti-correlated or modestly to weakly anti-correlated with each other. For both Contest tasks (`Task1` and `Task2`), and each model sub-group, the `LogSpectralNorm` trend (at best) increases with increasing model quality (in *dis*agreement with SLT). For `Task1`, it increases when considering all models together; however, for `Task2`, it *decreases* when grouping all models, exhibiting a clear Simpson's paradox. Also, for both Contest tasks, and for each model subgroup, `Alpha` decreases with increasing model quality (in agreement with HT-SR theory). For `Task1`, it decreases when considering all models together; however, for `Task2`, it *increases* when aggregating all models, again exhibiting a clear Simpson's paradox.

That is: (1) for both `LogSpectralNorm` and `Alpha`, the `Task2` models exhibit a Simpson's paradox; (2) the predictions of `LogSpectralNorm` *disagree* with (SLT) theory, except when aggregated when a Simpson's paradox is present; and (3) the predictions of `Alpha` (and `LogSpectralNorm`) *agree* agree with (HT-SR) theory, except when aggregated when a Simpson's paradox is present.

**Shape parameters and hyperparameter variation.** The *Scale* based `LogSpectralNorm` can do well when the data are aggregated, but it does very poorly when the data are segmented by architecture type (here, depth). More generally, this metric predicts test accuracy with large architectural changes, e.g., depth. However, it may behave oppositely to that suggested by bounding theorems, meaning that it can be *anti-correlated* with test accuracies, when varying the regularization hyperparameters. This confirms unexplained observations made in a different setting (Jiang et al., 2019).

The *Shape* based `Alpha` from HT-SR theory is predictive of test accuracy as hyperparameters vary, with large-scale architectural changes held fixed. Our results are the first to demonstrate that `Alpha`—a *Shape* metric—predicts test accuracy, as hyperparameters $\theta$ are varied independently. Also, and importantly, the `Alpha` metric performs better for higher quality models; this is evident comparing results for the `Task2` 2xx versus 6xx models. For models with better test accuracies, the `Alpha` metric predicts them better. This is seen visually in Figure 5, and also in Table 5, which compares both linear and rank correlation metrics, for model sub-groups with a fixed number of layers $L$.

**Extracting scale and shape metrics from pre-trained DNN models.** While computing norm-based *Scale* metrics is straightforward, computing HT-based *Shape* metrics is much more subtle. From HT-SR theory (Martin & Mahoney, 2021), we want to fit the tail of ESD of layer weight matrices to a PL distribution, as in Eqn. (1). This is accomplished with the `WeightWatcher` tool) (wei,

2018). The parameters $x_{max}$ (the largest eigenvalue) and $\alpha$ (the PL exponent) in Eqn. (1) (below) have a natural interpretation in terms of the scale and shape of the PL distribution, so we interpret the corresponding `LogSpectralNorm` and `Alpha` as *Scale* and *Shape* DNN model quality metrics.

## 2 PRELIMINARIES AND RELATED WORK

**Predicting trends in model quality.**    There is a large body of (older (Bartlett, 1997) and more recent (Neyshabur et al., 2015; Bartlett et al., 2017; Arora et al., 2018)) work from SLT on providing upper bounds on generalization quality of models; and there is a smaller body of (older (Engel & den Broeck, 2001) and more recent (Zdeborová & Krzakala, 2016; Martin & Mahoney, 2021; Bahri et al., 2020)) work using ideas from statistical mechanics to predict performance of models. Most relevant for us are the recent results on "Predicting trends in the quality of state-of-the-art neural networks without access to training or testing data" (Martin et al., 2021) and "Fantastic generalization measures and where to find them" (Jiang et al., 2019). The former work (Martin et al., 2021) considered a very broad range of CV and NLP models, including nearly every publicly-available pre-trained model, totaling to hundreds of SOTA models; and it focused on metrics that perform well on SOTA, without any access to training and/or testing data. The latter work (Jiang et al., 2019) considered a larger number models drawn from a much narrower range of CV models; and it considered a broad range of metrics, most of which require access to the training and/or testing data.

The Contest used a casual measure, similar to that in Jiang et al. (2019), to evaluate all the models, for both tasks, *in aggregate*, as opposed to evaluating the individual sub-groups, as we do here. This causal metric is highly suspect. In particular, we point to Dziugaite et al. (2020), who used very different methods than ours to obtain conclusions consistent with ours. They noted that the kind of causal measures proposed by Jiang et al. (2019) can obscure failures and successes; and they argued that generalization studies should be evaluated using an aggregate measure of "distributional robustness." There is also work that attempts to identify reliable generalization metrics, but using different evaluation methods than ours (Dziugaite et al., 2020; Liao et al., 2018; Thomas et al., 2019).

None of the works above, however, have considered that Simpson's paradoxes can arise in this kind of analysis. Pearl has argued for some time that causal relationships can not be inferred directly from observational data *because* of the presence of potential Simpson's paradoxes (Pearl, 2009). Simpson's paradoxes often arise in social science and biomedicine, since understanding causal relationships is particularly important in those areas. To our knowledge, this is the first application of these ideas in this area. Importantly, we would not have been able to observe the tradeoff between *Scale* and *Shape* metrics if we had chosen to use an aggregate, or, worse, a causal measure, here.

**Models from a contest.**    The Contest (Jiang et al., 2020a;b) covered a rather narrow class of CV models, but it provided multiple versions of each, trained with different numbers of layers (depths) and regularization hyperparameters. The approximately 150 CV models were organized into two architectural types, *VGG-like models* and *Network-in-Network models*, each with several sub-groups of fixed depth (number of layers $L$), and trained with different batch sizes, dropout, and weight decay.

See Table 4 in Appendix B for a summary of the models and sub-groups. There are two model groups: `Task1` and `Task2`. In every model sub-group (each row in Table 4), all models have the same depth (number of layers). `Task1` contains 2 Conv2D widths in each subgroup, whereas in `Task2`, only the the regularization is varied. Briefly, the models are:

- `Task1` ('task1_v4"): 96 VGG-like models, trained on CIFAR10, with 4 subgroups having 24 models each: 0xx, 1xx, 2xx, 5xx, 6xx, and 7xx.
- `Task2` ("task2_v1"): 54 models, with stacked Dense layers, trained on SVHN, with 3 subgroups having 18 models each: 2xx, 6xx, 9xx, and 10xx.

## 3 EXTRACTING SCALE / SHAPE PARAMETERS FROM PRE-TRAINED MODELS

### 3.1 COMPUTING SCALE AND SHAPE PARAMETERS WITH NORMS AND PL EXPONENTS

For completeness, we have evaluated several quality metrics, including: two from SLT (including `LogSpectralNorm` and `LogFrobeniusNorm`) (Jiang et al., 2019; 2020a;b); two from statistical mechanics and HT-SR Theory (including `AlphaHat` and `LogAlphaShattenNorm`) (Martin & Mahoney, 2021; 2019; 2020); and two (`Alpha` and `QualityOfAlphaFit`) that we introduce in this paper. See Table 3 and the discussion in Appendix A for a summary of the best performing.

All metrics reported are averages over the individual layer-metrics, as computed using the publically available and open-source `WeightWatcher`tool (wei, 2018). Both computing the norm

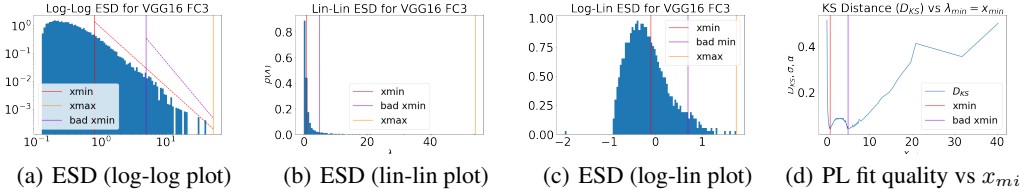

(a) ESD (log-log plot)    (b) ESD (lin-lin plot)    (c) ESD (log-lin plot)    (d) PL fit quality vs $x_{min}$

Figure 1: Illustration of the role of the ESD shape in determining the PL parameter $\alpha$ in the PL fit. Shown is VGG16, FC3 (a nearly "ideal" example). (See Appendix C for less ideal examples.)

metrics and fitting HT distributions has subtleties (Clauset et al., 2009; Newman, 2005), and `WeightWatcher` encapsulates the process, making it straightforward and reproducible.

**"Scale" parameters.** The `LogSpectralNorm` and `LogFrobeniusNorm` are layer averages over the logarithm of the corresponding norm (Jiang et al., 2019). (See Table 4 for details.) These norms have natural interpretation as a layer "scale" implicitly learned by the NN during training. The `LogSpectralNorm` is also measure of average HTness, and is anti-correlated with `Alpha` in many cases.

**"Shape" parameters.** The `Alpha` metric takes the layer average of the *PL exponent* $\alpha$, as fit to the tail of the layer ESD $\rho_{tail}(\lambda)$. The parameter $\alpha$ that is the same one that arises in the recently-developed HT-SR theory (Martin & Mahoney, 2021; 2019; 2020). The `QualityOfAlphaFit` metric is the average Goodness of Fit, determined using a Kolmogorov-Smirnov test (or KS-test), denoted $D_{KS}$. Neither `Alpha` nor `QualityOfAlphaFit` have been considered previously.

**Combining shape and scale.** The `AlphaHat` (Martin et al., 2021) and `LogAlphaShattenNorm` (Martin & Mahoney, 2020) combine Shape and Scale. `AlphaHat` which is a weighted average of layer $\alpha$, weighted by the layer $\log \lambda_{max}$. `LogAlphaShattenNorm` is logarithm of the Shatten norm of layer weight matrices, where the Shatten parameter for each layer is the $\alpha$ from the PL fit of that layer. The `AlphaHat` metric approximates the `LogAlphaShattenNorm` for the tail of the ESD, but both give similar results in practice.

### 3.2 FITTING ESDs TO PLs

In practice, these PL fits require some care to obtain consistent, reliable results; and, for this reason, here we describe some of these issues. Among other things, it is important to understand the behavior of the PL fit as $x_{min}$ is varied. See Figure 1 for an illustrative (very "nice") example; and see Appendix C for more discussion of these fitting issues, including dealing with non-ideal ESDs. (These figures can be generated by the `WeightWatcher` tool.)

In Figure 1, we consider the third fully connected (FC) layer from VGG16 (which was studied previously (Martin & Mahoney, 2021; Martin et al., 2021)), which is a nearly "ideal" example that we use to illustrate the method. In 1(a), 1(b), and 1(c), respectively, we show the ESD in log-log plot, linear-linear plot, and log-linear plot; and in each of those plots, we mark the position of $x_{max}$, the $x_{min}$ found by the fitting procedure, and a suboptimal value of $x_{min}$ chosen by hand.

The log-log plots, in 1(a) highlight the (well-known but often finicky) linear trend on a log-log plot. To aid the eye, in 1(a), we show the slopes on a log-log plot, as determined by the $\alpha$ fit for that value of $x_{min}$. The linear-linear plots, in 1(b), the usual so-called Scree plots, are not particularly informative. The log-linear plots, in 1(c), shows that the optimal value of $x_{min}$ is near the peak of the distribution, and that the distribution is clearly not log-normal, having a strong right-ward bias, with a spreading out of larger eigenvalues as $\lambda_{max}$ is approached.

In 1(d), we show the quality of the fit, measured by the KS distance, as a function of $x_{min}$. The interpretation of these PL fits is that, above that value of $x_{min}$ and below the value of $x_{max}$, the ESD is fit to a line, with slope $-\alpha$, in a log-log plot. In 1(d), we see that the suboptimal value of $x_{min}$ was chosen to be a local minimum in the the KS distance plot, and that the fit quality gradually degrades as $x_{min}$ increases. Looking at 1(c), we see that there is a very slight "shelf" in the ESD probability mass; and that the suboptimal value of $x_{min}$ corresponds to fitting a much smaller portion of the ESD

with a PL fit. For high-quality models, like this one, smaller values of $\alpha$ corresponds to better models, and decreasing $\alpha$ is well-correlated with increasing $\lambda_{max}$.[1]

## 3.3 COMPARING SCALE VERSUS SHAPE PARAMETERS

Here, we compare *Shape* versus *Scale* parameters, illustrating that they capture different information about models, as task, depth, and regularization / solver hyperparameters are varied.

See Figure 2, which compares `Alpha` and `LogSpectralNorm` for models from Table 4, segmented into subgroups corresponding to models with the same depth. See also Table 1 for more detailed numerical results. For some model sub-groups, `Alpha` and `LogSpectralNorm` are strongly anti-correlated; while for other subgroups, they are modestly to weakly anti-correlated, at best.

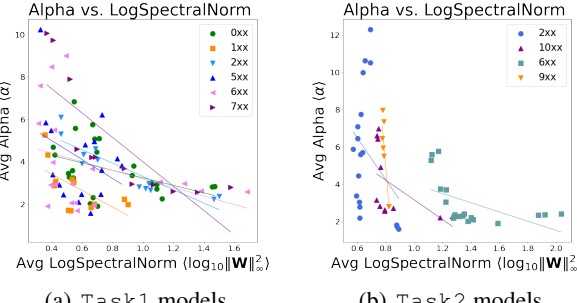

(a) `Task1` models.

(b) `Task2` models.

Figure 2: Comparison of the `LogSpectralNorm` and `Alpha` metrics, for `Task1` and `Task2` models.

For example, for `Task1`, in the $2xx$ and $9xx$ models, the two metrics are strongly anti-correlated metrics, with $R^2 > 0.6$ and a Kendall-$\tau$ rank correlation metric of $\tau > 0.75$;[2] in the $1xx$ models, they are modestly so, with $R^2 \sim 0.4$ and $\tau \sim 0.4$; and in the remaining model subgroups, $R^2 < 0.2$ and $\tau \leq 0.3$, which we identify as weakly correlated. Similarly, in `Task2`, we can identify model sub-group $9xx$ has having well anti-correlated average metrics, with large $R^2$ and Kendall-$\tau$; model sub-groups $6xx$ and $10xx$ exhibit intermediate behavior, as $R^2 < 0.3$ for each; and model sub-group $2xx$ shows no substantial correlation at all between `Alpha` and `LogSpectralNorm`. See Appendix D for more details on two illustrative pairs of examples.

|  | $R^2$ | Kendall-$\tau$ | Correlation |
|---|---|---|---|
| `Task1- 0xx` | 0.162 | 0.29 | Weak |
| `Task1- 1xx` | 0.405 | 0.394 | Modest |
| `Task1- 2xx` | 0.803 | 0.788 | Strong |
| `Task1- 5xx` | 0.124 | 0.117 | Weak |
| `Task1- 6xx` | 0.124 | 0.263 | Weak |
| `Task1- 7xx` | 0.64 | 0.909 | Strong |
| `Task1- AVG` | 0.38 | 0.46 | |
| `Task2- 2xx` | 0.113 | 0.0327 | None |
| `Task2- 6xx` | 0.282 | 0.451 | Modest |
| `Task2- 9xx` | 0.754 | 0.600 | Strong |
| `Task2- 10xx` | 0.273 | 0.636 | Modest |
| `Task2- AVG` | 0.36 | 0.43 | |

Table 1: Comparison of the `Alpha` and `LogSpectralNorm` metrics, for `Task1` and `Task2` models. $R^2$ and Kendall-$\tau$ for `Task1` and `Task2`, both aggregated and partitioned into model sub-groups.

## 4 A SIMPSON'S PARADOX: ARCHITECTURE VERSUS SOLVER KNOBS

### 4.1 BASIC PROPERTIES OF THE DATA

The obvious `Baseline` for model quality is the training accuracy. If we have access only to pre-trained models and no data (as in prior work (Martin et al., 2021) and as is assumed with `Alpha` and `LogSpectralNorm`), then we cannot check this baseline. Similarly, if the training error is *exactly* zero, then this is not a useful baseline. Otherwise, we can check against it, and we expect testing accuracy to improve as training accuracy improves.

In a practical setting, for a properly trained model, one expects that test accuracy will increase with increasing training accuracy, or at least not decrease, when the model capacity and dataset it fixed.[3] If the test accuracy does decrease, this indicates that the model may be somewhat

---

[1] This is predicted by HT-SR Theory (Martin & Mahoney, 2021; Martin et al., 2021), but it can be surprising from the perspective of SLT (Jiang et al., 2019; 2020a;b).

[2] Likewise, a strong linear correlation, not just good rank correlation, is predicted for good models by HT-SR theory (Martin & Mahoney, 2021), and this has been observed previously for SOTA CV and NLP models (Martin et al., 2021).

[3] When the model capacity or data set size is changed, then test accuracy may not track the training accuracy due to the Double Descent phenomena (Belkin et al., 2019), and the fact that NNs can exhibit complex phase behavior (Martin & Mahoney, 2017). While unlikely, we can not totally discount this for the `Task1` models.

overtrained and therefore will not generalize well. To examine this, we look at different model-groups which have a fixed depth ($L$). Figure 3 plots the relationship between training and testing accuracies for the `Task1` and `Task2` models, color-coded by model_number (0xx, 1xx, ...).

See also Table 2 for more detailed numerical results. By looking at the both the trendlines and the distribution of points in Figure 3, we can identify such overtrained models here: almost all of the `Task1` models (except perhaps the very highly accurate 5xx and 6xx models, and the moderately accurate 0xx and 1xx models); the `Task2` 9xx model; and to a lesser extent the `Task2` 2xx model.

For both `Task1` and `Task2`, there is a very large gap between the training and testing accuracies. A positive correlation in Figure 3 indicates that improving the training accuracy, even marginally, would lead to improved testing accuracy. Instead, we see that for `Task1`, for most model sub-groups (1xx, 2xx, 6xx, and 7xx),

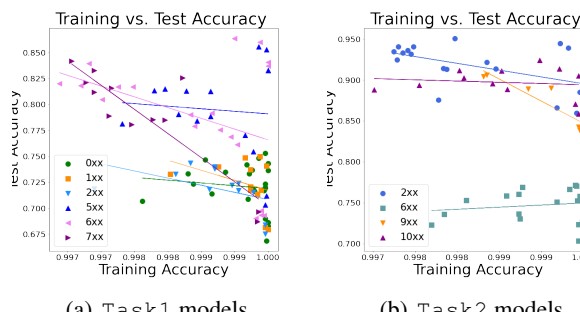

(a) `Task1` models.  (b) `Task2` models.

Figure 3: Relationship between training accuracy and testing accuracy for `Task1` and `Task2` models. One expects a positive correlation or (if the training error is near zero) at least not a negative correlation. Observe, however, that in many cases they are strongly anti-correlated. See also Table 2.

training and testing accuracies are (very) anti-correlated. For other two (0xx and 5xx), the trendline is only weakly anti-correlated. For all models, however, as the training accuracy approaches 1, the test accuracy plummets (except for a few outliers in the 5xx and 6xx groups). In this sense, these `Task1` models are over-trained; they clump into two sub-groups, those for this $R^2 \approx 0$ and those for which $R^2 > 0.1$. Similarly, for `Task2`, for model_number 9xx, they are (very) anti-correlated, and otherwise they are (slightly) anti-correlated for 2xx. The other two models show essentially no discernible trend. In all cases, improving training accuracy leads either to no noticeable improvement in test accuracy, or to worse testing performance.

Overall, many of the `Task1` models, as well as model `Task2` 9xx, behave qualitatively and quantitatively differently than the reasonably well-trained `Task2` models. The `Task1` models are generally of lower quality. Some interpolate exactly to $0$ test error while others do not—a significant difference. We suspect some may be overtrained. Notably, only the `Task2` models display the Simpson's paradoxes we identify here. It is those where we can understand why the `AlphaHat` metric works.

| | $R^2$ | RMSE | Kendall-$\tau$ | Correlation |
|---|---|---|---|---|
| `Task1` 0xx | 0.01 | 0.02 | 0.10 | Weak |
| `Task1` 1xx | 0.12 | 0.02 | 0.33 | Modest |
| `Task1` 2xx | 0.22 | 0.02 | 0.54 | Strong |
| `Task1` 5xx | 0.01 | 0.04 | 0.01 | Weak |
| `Task1` 6xx | 0.16 | 0.05 | 0.31 | Modest |
| `Task1` 7xx | 0.78 | 0.02 | 0.60 | Strong |
| `Task1` AVG | 0.22 | 0.03 | 0.31 | |
| `Task2` 2xx | 0.24 | 0.02 | 0.25 | Modest |
| `Task2` 6xx | 0.02 | 0.02 | -0.05 | Weak |
| `Task2` 9xx | 0.83 | 0.01 | 0.73 | Strong |
| `Task2` 10xx | 0.02 | 0.02 | 0.02 | Weak |
| `Task2` AVG | 0.28 | 0.02 | 0.24 | |

Table 2: Quality metrics (for $R^2$, larger is better; for RMSE, smaller is better; and for Kendall-$\tau$ rank correlation, larger magnitude is better) for the relationship between training error and testing error, as illustrated in Figure 3.

## 4.2 VISUALIZING THE SIMPSON'S PARADOX IN DNN MODELS

We now consider how the test accuracy varies against the `LogSpectralNorm` and `Alpha`, respectively, overall and broken down for each model sub-group, for all models from Table 4. See Figure 4 and Figure 5 for a summary. In our analysis, we first look at each model sub-group (0xx, 1xx, 2xx, ...) individually, which corresponds to a specific depth $L$, and we measure the regression and rank correlation metrics on the test accuracy as the hyperparameters vary.

From Figure 4, for each model sub-group, the `LogSpectralNorm` increases with increasing model quality. For `Task1`, the `LogSpectralNorm` increases when all models are considered together; but for `Task2`, the `LogSpectralNorm` *decreases* when all models are considered together, exhibiting a clear Simpson's paradox.

From Figure 5, for each model sub-group, the `Alpha` decreases with increasing model quality. For `Task1`, the `Alpha` decreases when all models are considered together; but for `Task2`, the `Alpha` *increases* when all models are considered together, exhibiting a clear Simpson's paradox.

Both metrics exhibit a clear Simpson's paradox for `Task2`.

Based on bounding theorems from SLT, one might expect that smaller values of `LogSpectralNorm` would correspond to better models. Similarly, based on HT-SR theory, one would expect that smaller values of `Alpha` would correspond to better models. With respect to these references, the `LogSpectralNorm` behaves the *opposite of what SLT would suggest, except* when considering the aggregated data (all models) when a Simpson's paradox is present (i.e., in `Task2`). On the other hand, the

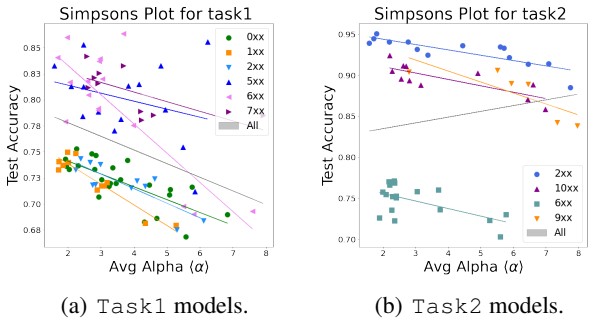

Figure 4: Test accuracy versus `LogSpectralNorm`, for `Task1` and `Task2`, overall and segmented by model sub-group. Observe the Simpson's paradox for `Task2` models.

(a) `Task1` models.   (b) `Task2` models.

Figure 5: Test accuracy versus `Alpha`, for `Task1` and `Task2`, overall and segmented by model sub-group. Observe the Simpson's paradox, in particular for `Task2` models.

`Alpha` behaves *precisely as what the HT-SR theory predicts, except* when considering the aggregated data when a Simpson's paradox is present. As shown below, however, by combining both into the `AlphaHat` metric, the Simpson's paradox in `Alpha` for the `Task2` models can be alleviated.

### 4.3 CHANGING ARCHITECTURES VERSUS CHANGING SOLVER HYPERPARAMETERS

Here, we provide a more detailed analysis of the main results presented in Figure 4 and Figure 5. See Figure 6 for histograms summarizing some of these results, in particular for `Alpha` and `LogSpectralNorm`. See also Table 5 and Table 6 in Appendix E for detailed statistics, including $R^2$ and Kendall-$\tau$ statistics, for metrics from Table 3.

From these, we see that `Alpha` (the mean PL exponent, averaged over all layers, which corresponds to a *Shape* parameter) *is correlated with the test accuracy for each DNN, when changing just the hyperparameters, $\theta$*. While it does not always exhibit the strongest correlation, it is the most consistent. We also see that `LogSpectralNorm` (the mean $\log_{10}$ spectral norm, also averaged over all layers, which corresponds to a *Scale* parameter) *is correlated with the test accuracy, when changing the number of layers, but it is anti-correlated when changing the hyperparameters*. Thus, it performs quite poorly when trying to identify more fine-scale structure.

For completeness, we have also included `LogFrobeniusNorm`, showing that it is often but not always correlated with test accuracy, and `QualityOfAlphaFit` (the KS distance), showing that it is correlated with test accuracy in about half the cases.

Finally, consider Figure 7, which shows the test accuracy versus `AlphaHat`, for `Task1` and `Task2`, segmented by model sub-group, and compare this with Figure 4 (for `LogSpectralNorm`) and Figure 5 (for `Alpha`). Recall that `AlphaHat` may be viewed either as a weighted `Alpha` (weighted by the layer $\lambda_l^{max}$) or as a weighted `LogSpectralNorm` (weighted by the layer $\alpha_l$).

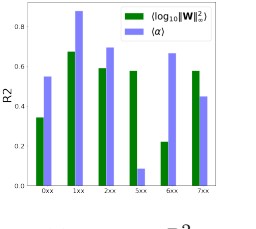 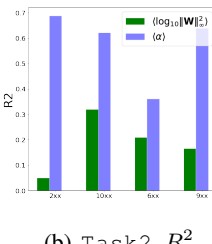 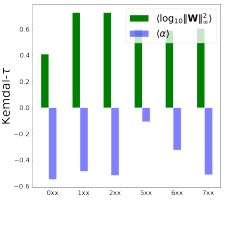 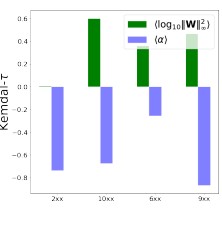

(a) `Task1`, $R^2$      (b) `Task2`, $R^2$      (c) `Task1`, Kendall-$\tau$      (d) `Task2`, Kendall-$\tau$

Figure 6: Comparison of predictions for test accuracies by `Alpha` $\langle \alpha \rangle$ and `LogSpectralNorm` $\langle \log_{10} \|\mathbf{W}\|_2^2 \rangle$, as measured using $R^2$ and Kendall-$\tau$ correlation and rank correlation metrics, respectively, for `Task1` and `Task2`, for each model sub-group. (See Table 5 for more details.)

As a weighted `Alpha`, the spectral norm weighting corrects for the fact that `Alpha` is scale-invariant, accounting for the variation in the scale of each weight matrix across different layers. As a weighted `LogSpectralNorm`, the $\alpha$ weighting corrects for the fact that the `LogSpectralNorm` is anti-correlated with the test accuracy when varying the regularization hyperparameters.

In either case, this combination "corrects for" the Simpson's paradoxes observed in Figures 4 and 5. This is the explanation for the previously-observed success of this metric at predicting the quality of SOTA DNN models (Martin et al., 2021).

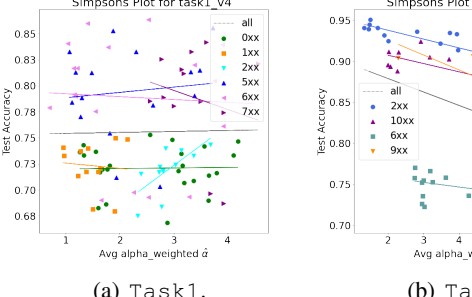

(a) `Task1`.      (b) `Task2`.

Figure 7: Test accuracy versus `AlphaHat`, for `Task1` and `Task2`, segmented by model sub-group. (Similar results are seen for `LogAlphaShattenNorm`.)

## 5 CONCLUSION

A goal of the HT-SR theory is to develop model quality metrics, like `Alpha` and `AlphaHat`, that can be applied to so-called pre-trained DNN models, by only watching the weights, i.e., without the need for access to training/testing data; and, in doing so, to also provide layer-by-layer quality metrics. Previous work demonstrated the utility of the `AlphaHat` metric when applied to a wide range of large scale and production, pre-trained CV and NLP models (Martin et al., 2021).

Here, we seek to better understand why the `AlphaHat` model quality metric works so well. We also present, for the first time, evidence that the `Alpha` metric correlates well with changes in regularization hyperparameters, and, in fact, works better with better performing models.

To accomplish this, we evaluated the `AlphaHat` metric and its subcomponent metrics, `Alpha` and `LogSpectralNorm`, on a set of publicly-available pre-trained models made available from a recent machine learning contest aimed at understanding causes of good generalization. To our initial surprise, we identified a clear Simpson's paradox in the data. From our exploration of that, we discovered the complementary roles of *Scale* metrics versus *Shape* metrics in evaluating model quality. Overall, our analysis explains the success of the `WeightWatcherAlphaHat` metric; it combines *Scale* and *Shape* information (Martin et al., 2021), and also shed light on previously-observed peculiarities of norm-based metrics (Jiang et al., 2019). Our results also highlight the need to go beyond one-size-fits-all– especially causal metrics– to describe the performance of SOTA NNs.

Based on our findings, we expect that `LogSpectralNorm` (and related *Scale*-based metrics) can capture coarse model trends due to changes in depth / number of layers whereas `Alpha` (and related *Shape*-based metrics) can capture fine-scale model properties (e.g., changes in regularization and other hyperparameters, including batch size, step scale, etc.) more generally, providing data-free diagnostics for DNNs on layer-by-layer basis. (See Appendix F for some additional discussion.)

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

## A  METRICS CONSIDERED IN OUR ANALYSIS

As mentioned in Section 2, we considered a range of metrics in our analysis. See Table 3 for a summary of the best performing. This includes both *average PL metrics*, from HT-SR Theory, and *average log-norm metrics*, from SLT, as well as metrics that combine the two approaches. (Other metrics performed worse, exhibited similar qualitative trends, or required access to training and testing data.)

| Complexity Metric | Average | Ref. | Scale or Shape? | Need data? | Need initial weights? | Need GPUs? |
|---|---|---|---|---|---|---|
| LogSpectralNorm | $\langle \log_{10} \|\mathbf{W}\|_2^2 \rangle$ | (Jiang et al. (2019)) | Scale | No | No | No |
| LogFrobeniusNorm | $\langle \log_{10} \|\mathbf{W}\|_F^2 \rangle$ | (Jiang et al. (2019)) | Scale | No | No | No |
| Alpha | $\langle \alpha \rangle$ | (this paper) | Shape | No | No | No |
| QualityOfAlphaFit | $D_{KS}$ | (this paper) | Shape | No | No | No |
| AlphaHat | $\hat{\alpha}$ | (Martin et al. (2021)) | Both | No | No | No |
| LogAlphaShattenNorm | $\langle \log_{10} \|\mathbf{W}\|_{2\alpha}^{2\alpha} \rangle$ | (Martin et al. (2021)) | Both | No | No | No |

Table 3: Overview of model quality metrics. Based on our initial analysis of Contest models, we propose and evaluate `Alpha` and `QualityOfAlphaFit`. The metrics in this table do not need access to training/testing data; they do not need information such as the initial weight distribution; and they do not require training/retraining (and thus access to GPUs).

**Average Power-Law (PL) metrics.**  Given a (pre-)trained DNN with $L$ layer weight matrices $\mathbf{W}$, PL metrics are computed by fitting the ESD of the correlation matrix $\mathbf{X} = \mathbf{W}^T\mathbf{W}$ of each layer to a PL, and then averaging over all layers.[4] In more detail, the tail of each layer ESD is fit to a PL of the form:

$$\rho_{tail}(\lambda) \sim \lambda^{-\alpha}, \ \ x_{min} \le \lambda \le x_{max}.$$

The fitting procedure selects the optimal PL exponent $\alpha$, and the adjustable parameter $x_{min}$. The PL exponent $\alpha$ characterizes the tail of the ESD; it measures what may be interpreted as the "shape" of most important part of the spectrum.

- `Alpha`: $\langle \alpha \rangle = \frac{1}{L} \sum_l \alpha_l$. This is a simple average of fitted $\alpha$ over all layers $L$. From the perspective of statistical mechanics, `Alpha` quantifies amount of correlation in the layer weight matrices.[5] From the perspective of statistics, `Alpha` may be viewed a shape parameter.
- `QualityOfAlphaFit`: $\langle D_{KS} \rangle$. For the given set of parameters $(\alpha, x_{min}, x_{max})$, the quality of the PL fit can be measured by the Kolmogorov-Smirnov (KS) distance ($D_{KS}$) between the empirical and theoretical distributions. The smaller $D_{KS}$ is, the better the fit.

We describe the PL fitting procedure in more detail (in Section 3.1), and we give examples of both high and low quality fits (in Appendix C). To our knowledge, we are the first to use `Alpha` and `QualityOfAlphaFit` to gauge model complexity.[6]

**Average Log-Norm metrics.**  Log-Norm metrics are related to *product-norm measures* of the model complexity $\mathcal{C}$. Given a (pre-)trained DNN with $L$ layers, and layer weight matrices $\mathbf{W}_l$, we define the $\mathcal{C}$ as the product over a norm of the layer weight matrices

$$\mathcal{C} := \|\mathbf{W}_1\| \times \|\mathbf{W}_2\| \times \cdots \|\mathbf{W}_L\|, \tag{3}$$

---

[4]For Conv2D($k, k, N, M$) layers, we extract $N = k \times k$ matrices of shape $N \times M$; typically, $k = 1$ or 3.

[5]Prior work (Martin & Mahoney, 2021) has argued that NNs resemble the strongly correlated systems, e.g., in electronic structure theory and statistical mechanics, which was the origin of early work in heavy-tailed random matrix theory (Bouchaud & Potters, 2003).

[6]Prior work (Martin et al., 2021) used a weighted version of this metric (AlphaHat). There, PL exponents $\alpha$ were computed, but they were not evaluated as a measure of test accuracy, and they were not shown to correlate with variations in solver hyperparameters (such as batch size, dropout, weight decay, etc.), as we do for the first time here.

where $\|\mathbf{W}_l\|$ denotes some arbitrary matrix norm for layer $l$.[7] If we take the logarithm of both side, we can express this complexity as an average over all layers ($L$), in log-units

$$\log_{10} \mathcal{C} \sim \langle \log_{10} \|\mathbf{W}\| \rangle := \frac{1}{L} \sum_{l=1}^{N} \log_{10} \|\mathbf{W}_l\|. \tag{4}$$

Metrics of this form provide a measure of the "scale" (in log units) of a model.

- `LogSpectralNorm`: $\langle \log_{10} \|\mathbf{W}\|_2^2 \rangle$. This is the (average) of the log of the layer Spectral norms. The layer Spectral Norm is just maximum eigenvalue $\lambda^{max}$ of the correlation matrix $\mathbf{X}$.[8]
- `LogFrobeniusNorm`: $\langle \log_{10} \|\mathbf{W}\|_F^2 \rangle$. This is the average of the log of the Frobenius norm for each layer weight matrix. It is included for completeness.

Importantly, these metrics take a layer average, and not a sum, since otherwise these metrics will trivially depend on the depth $L$ of the network.

**Combining PL and Log-Norm metrics.** Previous work considered metrics that combine "scale" and "shape" ideas (Martin & Mahoney, 2020; Martin et al., 2021), two of which we consider here.

- `AlphaHat`: $\hat{\alpha} = \frac{1}{L} \sum_l \alpha_l \log_{10} \lambda_l^{max}$. This has been used previously as a complexity measure for a large number of pre-trained NNs of varying depth and hyperparameters (Martin & Mahoney, 2020; Martin et al., 2021). `AlphaHat` may be viewed in one of two complementary ways: either as a weighted `Alpha` (weighted by the layer $\lambda_l^{max}$); or, equivalently, as a weighted `LogSpectralNorm` (weighted by the layer $\alpha_l$). As a weighted `Alpha`, the spectral norm weighting corrects for the fact that `Alpha` is scale-invariant, accounting for the variation in the scale of each weight matrix across different layers. As a weighted `LogSpectralNorm`, the $\alpha$ weighting corrects for the fact that the `LogSpectralNorm` is anti-correlated with the test accuracy when varying the regularization hyperparameters.
- `LogAlphaShattenNorm`: $\langle \log_{10} \|\mathbf{W}\|_{2\alpha}^{2\alpha} \rangle = \langle \log_{10} \|\mathbf{X}\|_{\alpha}^{\alpha} \rangle$. This has been used previously (Martin & Mahoney, 2020), and it is the average of the Log of the standard Shatten norm, defined with the value of $2\alpha$ from the PL fit of the ESD of each layer weight matrix $\mathbf{W}$. The `AlphaHat` metric approximates the `LogAlphaShattenNorm` under certain conditions (Martin & Mahoney, 2020).

Table 3 provides a summary of the metrics we considered. There are many other metrics that we examined but that we do not describe here: the path norm and Fisher-Rao metrics (Jiang et al., 2020a;b) are more expensive, perform worse, and/or don't add new insight; the Jacobian Norm is also too expensive to compute; and other metrics described in the Contest (Jiang et al., 2020a;b) either were uninteresting (e.g., the sum—not average—of layer norms, which is a proxy for depth) or performed very poorly.[9]

## B   MODELS CONSIDERED IN OUR ANALYSIS

See Table 4 for a summary of the models we considered in our analysis. These are described in more detail in Section 2. See also Jiang et al. (2020a;b) for more details.

## C   MORE ON DETERMINING SHAPE PARAMETERS (FITTING ESDS TO PLS), FROM SECTION 3.2

Fitting data to PLs is very finicky (Newman, 2005; Sornette, 2007; Clauset et al., 2009; Beggs & Timme, 2012; Alstott et al., 2014; Marshall et al., 2005). We have found it best to proceed with a combination of visual inspection and analysis with the `WeightWatcher` tool (wei, 2018).

---

[7]We drop the layer subscript $l$ when it is clear from the context.

[8]Prior work has shown that using norm-based metrics in log-scale tends to be superior to working with them in non-log scale (Martin et al., 2021). Even when taking averages, however, norm-based metrics, unlike `Alpha`, are *not* scale invariant.

[9]For the Conv2D layers, extracting the layer weight matrices some arbitrary choices; see Martin & Mahoney (2020) for a discussion of how the `WeightWatcher` tool does this

| Series | # | L | Batch Size | Dropout | Weight Decay | Conv Width | Dense | (k) |
|---|---|---|---|---|---|---|---|---|
| Task1 | 0xx | 4 | 8, 32, 512 | 0.0, 0.5 | 0.0, 0.001 | 256, 512 | 1 | 1 |
| "task1_v4" | 1xx | 5 | 8, 32, 512 | 0.0, 0.5 | 0.001 | 256, 512 | 2 | 1 |
| (VGG-like) | 2xx | 5 | 8, 32, 512 | 0.0, 0.5 | 0.0 | 256, 512 | 2 | 1 |
| | 5xx | 8 | 8, 32, 512 | 0.0, 0.5 | 0.001 | 256, 512 | 1 | 3 |
| | 6xx | 8 | 8, 32, 512 | 0.0, 0.5 | 0.0 | 256, 512 | 2 | 3 |
| | 7xx | 9 | 8, 32, 512 | 0.0, 0.5 | 0.0 | 256, 512 | 2 | 3 |
| Task2 | 2xx | 13 | 32, 512, 1024 | 0.0, 0.25, 0.5 | 0.0, 0.001 | 512 | - | - |
| "task2_v1" | 6xx | 7 | 32, 512, 1024 | 0.0, 0.25, 0.5 | 0.0, 0.001 | 512 | - | - |
| (Network- | 9xx | 10 | 1024 | 0.0, 0.25, 0.5 | 0.0, 0.001 | 512 | - | - |
| in-Network) | 10xx | 10 | 32, 512 | 0.0, 0.25, 0.5 | 0.0, 0.001 | 512 | - | - |

Table 4: Overview of models we considered in each task and sub-group, including variations in depth (L), regularization hyperparameters (Batch Size, amount of Dropout, and Weight Decay), and architectural changes (Width of selected Convolutional Layers, number of selected Dense layers, and kernel-size ($k = 1, 3$) for selected Convolutional layers). See Jiang et al. (2020a;b) for complete details.

**Visual inspection of ESDs.**   The following advice, taken directly from Sornette (Sornette, 2007), is particularly helpful for visual inspection of ESDs: "we recommend a preliminary visual exploration by plotting the survival and density distributions in (i) linear-linear coordinates, (ii) log-linear coordinates (linear abscissa and logarithmic ordinate) and (iii) log-log coordinates (logarithmic abscissa and logarithmic ordinate). The visual comparison between these three plots provides a fast and intuitive view of the nature of the data.

- A power law distribution will appear as a convex curve in the linear-linear and log-linear plots and as a straight line in the log-log plot.

- A Gaussian distribution will appear as a bell-shaped curve in the linear-linear plot, as an inverted parabola in the log-linear plot and as strongly concave sharply falling curve in the log-log plot.

- An exponential distribution will appear as a convex curve in the linear-linear plot, as a straight line in the log-linear plot and as a concave curve in the log-logp lot.

Having in mind the shape of these three reference distributions in these three representations provides fast and useful reference points to classify the unknown distribution under study." It is helpful to examine ESDs such as those in Figure 1 (in Section 3.2) and Figure 8 (here) in light of these comments.

Of course, such a visual inspection is just a first step to a more detailed analysis, since by itself visual analysis of HT data can be misleading. Sornette (Sornette, 2007) goes on to say: "While we recommend a first visual inspection, it is only a first indication, not a proof. It is a necessary step to convince oneself (and the reviewers and journal editors) but certainly not a sufficient condition. It is a standard rule of thumb that a power law scaling is thought to be meaningful if it holds over at least two to three decades on both axes and is bracketed by deviations on both sides whose origins can be understood (for instance, due to insufficient sampling and/or finite-size effects)." It is for these reasons that understanding the behavior of the ESDs near $x_{min}$ and $x_{max}$ (of Eqn. (1)) is so important.

**Fitting to non-ideal ESDs.**   In Figure 1, we illustrated how PL fits of ESDs perform on a nearly "ideal" example. Here, we discuss how it performs on less-than-ideal examples (that occurred in the Contest data). See Figure 8.

In Figure 8, we see several examples of layers that are less well-fit by a PL. In these cases, we see that the linear-linear plots are non-informative; the log-linear plots show that the distributions have a strong left-ward bias, indicating a tail of increasingly small eigenvalues; and there is not a broad range of large eigenvalues on the right of the distribution. This relative paucity of large eigenvalues is seen on the log-linear plots by a (more or less aggressive) truncation in probability mass for larger eigenvalues, and on the log-log plots by a steeper downward slope on the right of the ESD. (In these cases, compare with Figure 1.) The first row (Task1, model 152, layer 1) illustrates an incompletely-developed right tail (alternatively a "bulk-plus-spikes" model may be

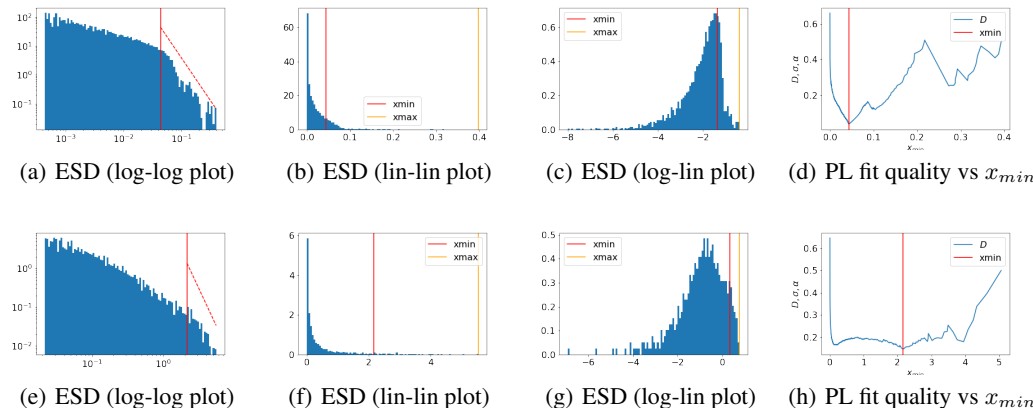

Figure 8: Illustration of the role of the ESD shape in determining the PL parameter $\alpha$ in the PL fit. Rows correspond to different layers in different models. Columns correspond to viewing the same ESD in different ways—on a log-log plot (first column), a linear-linear plot (second column), and a log-linear plot (third)—and the KS Distance $D$ of the PL fit as a function of $x_{min}$ (final column). Row 1: Task1, model 152, layer 1. Row 2: Task2, model 1006, layer 10.

a more appropriate fit than a PL fit), meaning that the spectral norm is somewhat smaller and that the fitting procedure has difficulty choosing $x_{min}$ near the peak of the distribution. The second row (Task2, model 1006, layer 10) illustrates an aggressively-shortened (effectively truncated, i.e., not even spikes) right tail, which leads to a much smaller spectral norm ($x_{max}$) and thus a much larger $x_{min}$ (since there is such a small range over which a linear fit is appropriate), as well as a broad range of (large) $x_{min}$ values over which low-quality fits are obtained. In each of these of these cases, the KS distance plots have less of a well-defined minimum as a function of $x_{min}$.

**Additional discussion.** In simple cases, scale and shape parameters do capture similar information. For models whose ESDs are very well-approximated by a Marchenko-Pastur (MP) distribution or a MP bulk-plus-spike distribution (RANDOM-LIKE and BULK+SPIKES phases from Martin & Mahoney (2021)), visual inspection of ESD plots often yields insight, and there is a strong correspondence between norm-based scale metrics and random MP-based shape metrics. Similarly, for models whose ESDs are very well-approximated by PL distributions (HEAVY-TAILED phase from Martin & Mahoney (2021), where the $x_{max} = \lambda_{max}$ truncation is not significant (Martin & Mahoney, 2021) to the fit), there is a strong correspondence between norm-based scale metrics and PL-based shape metrics, in the sense that smaller $\alpha$ values correspond closely to larger $\lambda_{max}$ values. For realistic models, however, and for the models from Table 4, these two metrics can be very different and can reveal very different information.

The broad range of behavior seen in the Contest data arises since ESDs look like ones in the rows of Figure 8 (rather than the more "ideal" case shown in Figure 1) where the linear fit to the ESD on log-scale is not very good. For models where Alpha against LogSpectralNorm behave more similarly, the ESDs look more like Figure 1. For the Contest models shown in Figure 8 (and others), for a simple (non-truncated, long tail) PL distribution, smaller exponents $\alpha$ correspond to a longer tail, which corresponds to a larger $\lambda_{max}$. However, since the tail of the ESDs are typically best fit by a PL—often with exponents $\alpha$ much larger than expected, since many of the models from Table 4 are of lower quality—the situation is not so simple. This simple connection only holds for a few model sub-groups. Generally speaking, these scale and shape metrics (LogSpectralNorm and Alpha) characterize the model ESDs differently and capture different properties of the models.

# D ILLUSTRATIVE EXAMPLES: COMPARING SCALE VERSUS SHAPE PARAMETERS

We saw in Figure 2 a comparison of the Alpha and LogSpectralNorm metrics, for Task1 and Task2 models. To get more detailed insight, Figure 9 plots Alpha versus LogSpectralNorm for

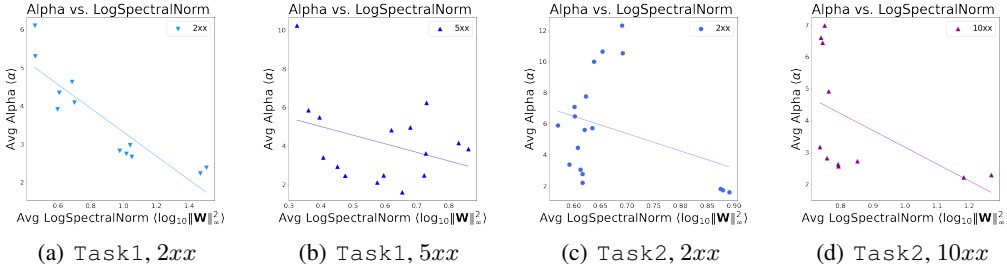

| (a) Task1, $2xx$ | (b) Task1, $5xx$ | (c) Task2, $2xx$ | (d) Task2, $10xx$ |

Figure 9: Comparison of Alpha and LogSpectralNorm for selected model sub-groups from both tasks. Lines (simply to guide the eye) shows a linear regression on the data.

two illustrative pairs of examples, from each of Task1 and Task2. Consider, as a baseline example, Task1, model sub-group $2xx$, in Figure 9(a). Here, the two metrics are strongly anti-correlated, with linear correlation metric $R^2 = 0.803$ and Kendall-$\tau$ rank correlation metric $\tau = 0.788$. In contrast, for Task1, model sub-group $5xx$, shown in 9(b), the two metrics are (at best) only weakly correlated, with $R^2 = 0.124$ and $\tau = 0.177$. This is typical; some model sub-groups exhibit large rank and/or linear correlations, others virtually none at all. For Task2, model sub-group $2xx$, versus Task2, model sub-group $10xx$, shown in 9(c) and 9(d), we see an example where the two plots look similar visually, and both have small(ish) linear correlation $R^2$, but they have very different Kendall-$\tau$ rank correlation metrics.

## E  ADDITIONAL DETAILS ON CHANGING ARCHITECTURES VERSUS CHANGING SOLVER HYPERPARAMETERS

|  | LogSpectralNorm | | LogFrobeniusNorm | | Alpha | |
|---|---|---|---|---|---|---|
|  | $R^2$ | Kendall-$\tau$ | $R^2$ | Kendall-$\tau$ | $R^2$ | Kendall-$\tau$ |
| Task1- 0xx | 0.34 | 0.41 | 0.00 | -0.03 | **0.55** | **-0.55** |
| Task1- 1xx | 0.68 | **0.73** | 0.29 | -0.30 | **0.88** | -0.48 |
| Task1- 2xx | 0.59 | 0.73 | 0.42 | 0.55 | **0.69** | -0.52 |
| Task1- 5xx | **0.58** | **0.60** | 0.19 | -0.25 | 0.09 | -0.10 |
| Task1- 6xx | 0.22 | **0.59** | 0.05 | -0.10 | **0.67** | -0.32 |
| Task1- 7xx | 0.58 | **0.61** | 0.38 | 0.58 | 0.45 | -0.51 |
| Task1- AVG | 0.50 | **0.61** | 0.22 | 0.07 | **0.55** | -0.41 |
| Task2- 2xx | 0.05 | 0.01 | 0.67 | **-0.84** | 0.69 | -0.74 |
| Task2- 10xx | 0.32 | 0.60 | 0.60 | -0.53 | 0.62 | **-0.67** |
| Task2- 6xx | 0.21 | **0.36** | 0.01 | -0.05 | **0.36** | -0.25 |
| Task2- 9xx | 0.17 | 0.47 | 0.65 | **-0.87** | 0.64 | -0.87 |
| Task2- AVG | 0.19 | 0.36 | 0.48 | -0.57 | 0.58 | -0.63 |

Table 5: Model quality for different metrics (all those mentioned in Table 3), for Task1 and Task2, both overall and by model sub-group. (This table is part 1 of 2; see also Table 6.)  Bar plots for Alpha and LogSpectralNorm data from here are shown in Figure 6.

See Table 5 and Table 6 for additional details on changing architectures versus changing solver hyperparameters, from Section 4.3.

## F  ADDITIONAL DISCUSSION

We conclude with a few more general thoughts on our results.

As is well known, trying to extract causality from correlation is difficult—precisely since there may be Simpson's-like paradoxes present in the data, depending on how the data are partitioned. When confronted with a Simpson's paradox, one is tempted to ask whether the marginal associations or

| | AlphaHat | | LogAlphaShattenNorm | | QualityOfAlphaFit | |
|---|---|---|---|---|---|---|
| | $R^2$ | Kendall-$\tau$ | $R^2$ | Kendall-$\tau$ | $R^2$ | Kendall-$\tau$ |
| Task1- 0xx | 0.00 | -0.06 | 0.00 | -0.02 | 0.02 | -0.10 |
| Task1- 1xx | 0.01 | 0.00 | 0.00 | -0.03 | 0.00 | 0.15 |
| Task1- 2xx | 0.49 | 0.70 | 0.48 | 0.61 | 0.67 | **-0.82** |
| Task1- 5xx | 0.01 | 0.10 | 0.00 | 0.08 | 0.04 | 0.18 |
| Task1- 6xx | 0.00 | -0.04 | 0.01 | -0.03 | 0.10 | -0.13 |
| Task1- 7xx | 0.04 | -0.06 | 0.07 | -0.03 | **0.60** | -0.58 |
| Task1- AVG | 0.09 | 0.11 | 0.09 | 0.10 | 0.24 | -0.22 |
| Task2- 2xx | 0.77 | -0.75 | 0.62 | -0.73 | **0.89** | -0.83 |
| Task2- 10xx | 0.41 | -0.38 | 0.46 | -0.42 | **0.70** | -0.67 |
| Task2- 6xx | 0.12 | -0.14 | 0.12 | -0.08 | 0.35 | -0.25 |
| Task2- 9xx | 0.59 | -0.87 | 0.65 | -0.87 | **0.91** | -0.87 |
| Task2- AVG | 0.47 | -0.53 | 0.46 | -0.53 | **0.71** | **-0.66** |

Table 6: Model quality for different metrics (all those mentioned in Table 3), for Task1 and Task2, both overall and by model sub-group. (This table is part 2 of 2; see also Table 5.) Bar plots for Alpha and LogSpectralNorm data from here are shown in Figure 6.

partial associations are correct. Often, the answer is that both are correct, depending on what is of interest. In particular, simple statistical analysis does not provide any guidance as to causal relationships and whether the marginal association or the partial association is the spurious relationship. For that reason, rather than trying to extract causality, we took a different approach: we looked for a Simpson's paradox; and when we found it, we tried to interpret it in terms of scale versus shape metrics from SLT and HT-SR theory.

One might wonder "why" LogSpectralNorm and Alpha perform as well as they do, when restricted to changes in the model depth and/or solver hyperparameters, respectively. Establishing such a causal explanation, of course, requires going beyond the data at hand and requires some sort of counterfactual analysis. This is beyond the scope of this paper. A plausible hypothesis, however, is the following. Since upper bounds from SLT suggest that models with smaller norms have less capacity, these norms are used (either explicitly as a regularizer, or implicitly by adjusting large matrix elements/columns/rows) during the training process, in particular when one varies coarse model parameters such as depth. On the other hand, coming from HT-SR theory, Alpha is not used explicitly or implicitly during the training process. Instead, the training process extracts correlations over many size scales from the data, and it is these correlations that are captured by smaller Alpha values, consistent with HT-SR theory and practice (Martin & Mahoney, 2021; 2019). This hypothesis is consistent with "why" fitted PL metrics from HT-SR theory—in particular the fitted AlphaHat metric–perform so well, both for the models considered in this contest, as well as for a much wider range of publicly-available SOTA DNN Models (Martin et al., 2021). Testing this hypothesis is an important question raised by our results.

