# OpenReview forum: "Post-mortem on a deep learning contest: a Simpson’s paradox and the complementary roles of scale metrics versus shape metrics"
_ICLR.cc/2023/Conference — Submitted to ICLR 2023_

### Official Review · Reviewer_tNyg · 2022-10-21

**Confidence:** 3
**Correctness:** 4
**Technical Novelty And Significance:** 2
**Empirical Novelty And Significance:** 2
**Recommendation:** 3

**Clarity, Quality, Novelty And Reproducibility:**

Clarity:
- the paper is sufficiently clear, although there are acronyms that are defined after their use (e.g., ESD), and graphs where the x-axis variables are not visible (e.g., Figure 1).

Quality:
- the analysis in the paper is purely empirical, and lacks theoretical justification on why alphahat (previously published in 2021) should do well.

Novelty:
- the materials in the paper is probably novel, but is publicly available and referenced (by the same authors) in their 2021 journal paper.

Reproducibility:
- the weight watcher code is publicly available, and so the implementations of the subcomponents should be in that code.



**Strength And Weaknesses:**

Strengths:
- The paper analyses the success of the alphahat metric, and showed that its two components separately capture scale and shape.

Weaknesses:
- The paper simply analyses the failures of alpha and the log spectral norm. There is not really any theoretical justification in this paper on why combining the two components in alphahat should result in its good performance.
- this submission has been referenced in [1] published in 2021. I think it could serve as an appendix to the journal paper, but the contribution is insufficient for a full paper at ICLR.


[1] Martin, Charles H., and Michael W. Mahoney. "Implicit Self-Regularization in Deep Neural Networks: Evidence from Random Matrix Theory and Implications for Learning." J. Mach. Learn. Res. 22.165 (2021): 1-73.

**Summary Of The Paper:**

The paper explains the success of the alphahat metric [1] in predicting model generalisation: the alphahat metric consists of two components, (i) alpha, the power law exponent in the empirical spectral density, and (ii) the log spectral norm. The paper shows that these components result in a Simpson's paradox in the analysis of their prediction of generalizability: the log spectral norm is scale based, and does well in aggregated data, while alpha is shape based, and does well in sub partitions of the data, but less well in aggregation. The authors argue that alphahat works well by combining these two components into a single metric.

[1] Martin, Charles H., and Michael W. Mahoney. "Implicit Self-Regularization in Deep Neural Networks: Evidence from Random Matrix Theory and Implications for Learning." J. Mach. Learn. Res. 22.165 (2021): 1-73.

**Summary Of The Review:**

I propose to reject the paper as
- the contribution as an empirical analysis of results published in 2021 does not qualify it for ICLR
- the paper has been referenced in 2021 by a publication by the same authors.

---

> ### Author Response · Authors · 2022-11-07
> **this paper uses previously published theory in a new use case; and the reviewer has a confusion about the order of appearance of previous papers**
>
> This work is indeed a direct follow-up to two papers, each of which was published in a top-tier journal (JMLR and Nature Communications):
>
> Implicit Self-Regularization in Deep Neural Networks: Evidence from Random Matrix Theory and Implications for Learning
> https://jmlr.org/papers/v22/20-410.html
>
> and
>
> Predicting trends in the quality of state-of-the-art neural networks without access to training or testing data
> https://www.nature.com/articles/s41467-021-24025-8
>
> You had two main concerns: first, that there is no theoretical justification to our approach; and second, that it should be an appendix to a paper that first appeared nearly three years earlier.
>
> Regarding the first point: it is simply not true that there is no current and elementary theoretical justification for the alpha-hat metric.  Perhaps we did not describe it well enough due to the space constraints for a conference paper, but that theory is clearly outlined in those previous papers which we cite.
>
> The TLDR of HTSR theory is that: the more heavy-tailed the layer ESDs are, the better the model will generalize; and, for the best-performing models, their layer ESDs will follow the expected limiting form from the universality class of heavy-tailed random matrices, namely, it will fit a power law.  This is predicted by theory outlined in previous papers that we cite.
>
> Because of this, we can usually fit the layer ESDs of well-trained models to a power law, with the layer exponent (alpha).
>
> To apply the HTSR theory to a large model, one can define a DNN model quality metric as an average of all the layer alphas.
>
> The  alpha-hat metric is simply a weighted average of the layer alpha metrics from the HTSR theory, where the weight adjustment simply corrects for the different scales of the weight matrices in each layer.  This is also described in prior work by Martin and Mahoney that we also cite (https://arxiv.org/abs/1901.08278).
>
> The layer-averaged alpha-hat metric has been extensively tested on hundreds of well-trained models and it does indeed appear to be well correlated with the reported test accuracies of these models.  In other words, the alpha-hat metric acts like a DNN model quality metric, with previously-established empirical and theoretical justification.
>
> In this paper, we seek to understand better when and why the alpha-hat metric performs well, and when it fails, so that among other things, we improve the formulation and application of the HTSR theory, and we did this by applying the theory to a corpus of data that appeared after the original theory was published.
>
> It was our understanding that empirical analysis of existing theoretically-principled quality/generalization metrics was of interest to the ICLR community.
> This is especially true when our analysis revealed a Simpson's paradox in a deep learning contest that received a fair amount of attention within the community and that was nominally about the \emph{causes} of generalization.
>
> Regarding your concern about this paper being cited: this submission first appeared nearly three years after the HTSR paper, and while it was obviously not cited in the original arXiv paper, it did appear in time for Martin and Mahoney to cite it in the final JMLR version of their HTSR paper.  It seems inappropriate for the authors of this paper to consider adding an appendix to a 73 page paper that first appeared nearly three years earlier and that analyzed data from a contest that appeared after the original HTSR paper was posted to the arXiv.
>
> We ask that you revise your comments and score.

---

### Official Review · Reviewer_9hRU · 2022-10-26

**Confidence:** 3
**Correctness:** 3
**Technical Novelty And Significance:** 2
**Empirical Novelty And Significance:** 2
**Recommendation:** 3

**Clarity, Quality, Novelty And Reproducibility:**

See under Weaknesses:
- The work is largely tied to particular ideas from a particular theory, and it feels like an addendum to a well-known existing work from specific researchers, so novelty is quite limited, and quite narrow.
- The paper is quite difficult to follow for non-experts, and it remains unclear what impact it will have on wider research beyond the HT-SR theory. It is also unclear if the HT-SR theory is really the right theory - I am not convinced by it given how the paper was motivated. The presentation should be improved.

**Strength And Weaknesses:**

Strengths:
- An atypical piece of (meta-)research that aims at an ambitious goal: analysing and comparing neural models without having access to training or test data, offering new insights on a particular phenomenon: Simpson's paradox.

Weaknesses:
- The whole work is basically an incremental piece of research heavily focused on one specific theory built in the work of Martin & Mahoney, and I am unsure how much impact this type of research will have on a wider ML audience. I am not convinced that the HT-SR theory is the right way to go in the first place when it comes to analysing and comparing different models. And it seems that the main goal of this particular work aims to improve one very specific metric (AlphaHat). The whole work seems a bit like a technical addendum to the source AlphaHat paper, as also stated by the authors themselves: "Most importantly, this work helps clarify why the AlphaHat metric performs so well across so many models."
- The work is also slightly detached from 'real-world' experiments: what consequences this type of research will have on ML models in practice and in production? What are the main take-home messages for researchers and practitioners working outside this niche area and not interested in in-vitro model analyses directly?
- The whole Contest (and the subsequent analyses) are based on 'simple/smaller-arhictecture) NNs in computer vision - the authors 'over-claim' that similar findings will hold for NLP models as well, but this is never empirically verified - they make conclusions 'by proxy', and I am unsure how much the findings hold across NNs with different architectures (e.g., CNN-based versus Transformer-based versus LSTM-based) and with different parameter budgets and depths. This again brings me to the question of relevance of this research.
- The presentation requires much more work - it is really difficult for someone who is not strictly in-domain (and not knowledgeable in the HT-SR) to follow the paper. First, the actual theory has to be defined with all the important preliminaries, the Simpson's paradox also needs a clear definition, and the Contest should also be described in more detail.

**Summary Of The Paper:**

Disclaimer: This is a rather unusual and unusually written (meta-)paper that falls outside my core area of expertise. I still tried to review the paper in good faith.

The work combines two lines of work: L1. first line aims to analyse generalisation capacity of neural networks (NNs) -- pretrained models -- directly without having access to any training or test data using and extending a particular metric from the recently-developed
Heavy-Tailed Self-Regularization (HT-SR) theory; L2. second line of work offered a contest with a series of pretrained Computer Vision models for a similar purpose (analysing and predicting generalisability). The idea is to run the basic framework from L1 on the available models from the L2 Contest and improve some mechanisms (e.g., the AlphaHat metric) typically used in L1.

The main contribution of this work is then the extension of the HT-SR theory which is at heart  of L1, based on the findings and analyses on the Contest (L2). The authors, among other things, show the disadvantage of the currently used single metric and decompose it into two metrics, and empirically validated the existence of a specific phenomenon (i.e., Simpson's paradox) in the Contest data.

**Summary Of The Review:**

While my review should be taken with a few grains of salt, my impression is that this work is too tied to a very narrow idea, and will result in very limited impact; the presentation should be reorganised, and the paper should also provide suggestions on how to connect its findings to more application-based ML (which it doesn't do at the moment): how can it inform our model selection in CV or NLP research?

---

> ### Author Response · Authors · 2022-11-07
> **atypical piece of (meta-)research that aims at an ambitious goal; and which seems incremental to those not familiar with the approach**
>
> Thank you for your good-faith comments, for acknowledging that this paper falls outside your core area of expertise, and for recognizing that this paper is an "atypical piece of (meta-)research that aims at an ambitious goal: analyzing and comparing neural models without having access to training or test data, offering new insights on a particular phenomenon: Simpson's paradox."
>
> You had several concerns, which we will address.
>
> 1. A main criticism of your is that the work is incremental.  We respectfully submit that this is due to your lack of expertise, as we will try to outline.
>
> 2. We agree with you that the work is "tied to particular ideas from a particular theory" and that it is "heavily focused on one specific theory built in the work of Martin & Mahoney."
> That is how a lot of research is, especially when the novel ideas work, as they do here.  We do not view that as a flaw.
> In particular, this submission is a followup to an existing line of prior art, in Martin, Peng and Mahoney (MPM21)(Nature Communications 2021: https://www.nature.com/articles/s41467-021-24025-8).
>
> 3. As to your comment, "I am not convinced that the HT-SR theory is the right way to go in the first place when it comes to analyzing and comparing different models."
> Aside from being a vague non-refutable feeling, this concern is irrelevant.  The reason it is irrelevant is because there is extensive prior art establishing the merits of this approach.
> Indeed, it was shown in MPM21 that the Alpha-Hat metric works remarkably well at "predicting trends in the quality of state-of-the-art neural networks without access to training or testing data."
>
> We should note that the Contest was announced after the original version of the MPM21 paper and the Martin-Mahoney JMLR paper on HT-SR theory.  HT-SR metrics were not included in the list of fantastic generalization metrics that were evaluated by the Contest, and thus it seemed appropriate to analyze the Contest data with HT-SR metrics.  So, yes, we agree, this work is a follow up, tied to a particular theoretical approach that the Contest failed to consider (since the Contest was tied to a different theoretical approach), and which intends to use the Contest data to better understand why the Alpha-Hat metric work so well, as prior work has clearly demonstrated.
>
> 4. With regard to your comment "The work is also slightly detached from 'real-world' experiments: what consequences this type of research will have on ML models in practice and in production?":
> We have to \emph{strongly} disagree.
> The HTSR theory has been implemented in an open-source tool, featured in the prior work MPM21 (see http://weightwatcher.ai).  This tool has been downloaded over 75K times, it has over 1000 stars, and it is currently in use in multiple production environments.
>
> 5. With regard to your comment "I am unsure how much the findings hold across NNs with different architectures (e.g., CNN-based versus Transformer-based versus LSTM-based) and with different parameter budgets and depths.":
> Again, the prior art (MPM21, Yang et al, etc.) has already shown the utility of the approach when applied across hundreds of different CV and NLP models.  That is not the focus of this paper.
>
> We agree that "the actual theory has to be defined with all the important preliminaries, the Simpson's paradox also needs a clear definition, and the Contest should also be described in more detail."  It is difficult to do everything in the the page limit of a conference submission, especially for an atypical piece of (meta-)research.  However, we are happy to make specific adjustments as possible.  We do note, however, that previous papers which we cite do describe all this material in detail:
> - The HTSR theory is described in Martin and Mahoney 2021 (https://jmlr.org/papers/v22/20-410.html)
> - Simpson's paradox is textbook material, and we refer to references in our paper (Simpson, 1951; Bickel et al., 1975; Robinson, 2009; Kievit et al., 2013).
> - the Contest is described in (Jiang et al., 2020a;b).
>
> We are happy to address other questions/concerns you have in light of our comments, and we ask you to raise your score in light of these comments.

---

### Official Review · Reviewer_WuLE · 2022-10-28

**Confidence:** 4
**Clarity, Quality, Novelty And Reproducibility:** Clarity
**Correctness:** 1
**Technical Novelty And Significance:** 2
**Empirical Novelty And Significance:** 2
**Recommendation:** 1

**Strength And Weaknesses:**

Strengths: understanding generalization is of great importance, the authors have some novel ideas here.

Weaknesses: the arguments made in the paper are extremely hard to follow. A serious round of editing and reorganization would greatly improve the readability of the paper.

**Summary Of The Paper:**

The paper describes a large number of experiments relating the AlphaHat measure to generalization performance of DNNs.

**Summary Of The Review:**

The goals of this work are important, and I think there are some interesting ideas here.

The writing is unfortunately very very hard to follow though. There are few technical details, and a great deal of opinion and prose.

As one example, take the end of page 2, the phrase "In investigating, why AlphaHat works..." The AlphaHat metric is defined just before this, but absolutely no context is given for what it would mean for the metric "to work". This kind of vagueness leads the reader guessing at many crucial points in the paper.

The paper goes on "Being aware of challenges with designing good contests and with extracting causality from correlation (Pearl, 2009), we did not use the causal metric provided by the Contest. Instead, we adopted a different approach: we identified Simpson’s paradoxes within the Contest data; and we used this to understand better the AlphaHat metric from HT-SR theory"

I found this impossible to follow. What was "the causal metric provided by the Contest"? It would be simple enough to describe this, it seems crucial to the general argument. What does it mean to identify Simpson's paradoxes in this setting? It's very hard for the reader to understand what is going on.

Overall I think there are probably some interesting ideas here, but I would strongly encourage the authors to better explicate their work, and the context for their work.

---

> ### Author Response · Authors · 2022-11-07
> **your two main concerns were addressed in the paper, and your criticisms are ill-founded, we think because you did not understand the paper**
>
> Thank you for your comments.  We will try to address each of them.
>
> To start, we are sorry the text is confusing to you, and we are happy to clarify any specific questions you have to update the paper accordingly.
> Other reviewers have not had such trouble reading the paper (one explicitly said "the paper is sufficiently clear"), so it is difficult for us to know where it is confusing to you and where it is not.
>
> Also, we are sorry you felt that there were few technical details, and we are happy to clarify any specific questions you have to update the paper accordingly.
> Again, though, other reviewers found the technical contribution compelling (for example, one explicitly said "it is a solid paper with extensive experiments for an important research question"), so again it is difficult to know the source of your confusion.
>
> On the other two specific issues raised:
>
> (1) what it would mean for the metric "to work".
>
> We refer you to the third sentence in the paper:
>
> "[the AlphaHat metric, (i.e., $\hat{\alpha}$), showing that it can predict trends in the quality, or generalization capacity, of state-of-the-art (SOTA) DNN models without access to any training or testing data (Martin et al., 2021)]"
>
> This means that, when looking across hundreds of pretrained models, and when grouped together appropriately, then the AlphaHat metric is predictive of the reported test accuracies.   That is, the metric shows the same trend as the quality of the model.  In addition, this metric is better than other existing metrics, as extensively demonstrated in the prior work.
> You criticized us for not providing context and being vague.  This is the context you say that we do not provide.  It is explicit in the third sentence of the paper; and it is not vague, as you incorrectly assert.
>
> (2)  What was "the causal metric provided by the Contest"?
>
> It is irrelevant.  The specific form is not crucial for our general argument, as you incorrectly assert.
>
> The reason why it is irrelevant is the following.  It is well known that one can not extract causal relationships directly from observational data, precisely because of the potential presence of Simpson's paradoxes. This is exactly why it is essential that we "grouped the data together appropriately" to observe the underlying trends.
>
> In our work, we show that if one simply groups the contest models by depth, then the HTSR alpha and alpha-hat metrics are well correlated with the reported test accuracies.  This is clearly depicted in Figure 3, and most evident for the very Task2 models.
> Again, you say that are main claims are not supported by empirical results.  That is false; see, e.g., Figure 3.
>
> We are presenting challenging but important results to the community.  We thank you for your efforts.  We ask that you revise your comments and your incorrect claim that our results "are incorrect or not at all supported by theory or empirical results."

---

### Official Review · Reviewer_E8e4 · 2022-10-31

**Confidence:** 3
**Correctness:** 3
**Technical Novelty And Significance:** 3
**Empirical Novelty And Significance:** 3
**Recommendation:** 6

**Clarity, Quality, Novelty And Reproducibility:**

Typo: "a *casual* measure"

I suggest the authors talk more about why the dataless assumption is used for this paper.

**Strength And Weaknesses:**


Strengths

The analysis is very extensive and provides an insightful and convincing explanation of why AlphaHat is powerful for predicting model quality. It has both extensive theoretical analysis and empirical results to support the hypothesis. The discussion on the Shape vs Scale fills the gap in the current literature. I particularly like the discussion on why the generalization across a broad range of CV and NLP models.

Weakness

I think the current presentation assumes that readers have enough background knowledge. Also, it will be better if the authors also introduce the motivation with real-world applications for illustration.



**Summary Of The Paper:**

This paper presents an analysis of why a particular model quality metric named AlphaHat works well. The AlphaHat metric is used to predict model performance without accessing the training and testing datasets. It only looks at the model weights and has two subcomponents, Alpha and LogSpectralNorm. The authors identify that there is a clear Simpson's paradox in the data and they found that Scale and Shape play two complementary roles as metrics for evaluating the model quality.


**Summary Of The Review:**

Overall, it is a solid paper with extensive experiments for an important research question. The presentation can be improved to make the paper more accessible to people who are not familiar with AlphaHat and the Contest. Using more figures with examples can make it better. The novelty is good and I think it presents useful insights for the community to continue this line of research.

---

> ### Author Response · Authors · 2022-11-05
> **on the dataless assumption**
>
> Thank you for your positive comments.  We want to address your question about "why the dataless assumption is used for this paper."
>
> By starting with a data-free theory, one is able to test the theory on the very large collection of open-source pre-trained DNN models, and, in particular, on the models that have been trained on the largest and most complete data sets and across domains of CV and NLP.   When this line of work started, there were fewer than 100 publicly-available pre-trained models available; there are now over 50K.  We argue that the data-free approach offers a unique approach for extensive meta-analysis on this massive trove of data.
>
> Moreover, having a data-free theory and the associated open-source weightwatcher tool has proven to be of great practical value in commercial settings to detect potential problems in production models where the training and even test data are difficult or impossible to obtain due to privacy, compliance, and/or other issues.
>
> Indeed, this approach, AFAIK, is the only approach that can detect potential problems in the specific layers of a trained DNN (and without needing data!).
>
> As to the direct practical value, please note that the weightwathcher tool (which is publicly-available and which we used) has over 75K downloads and over 1000 stars on GitHub.
>
> https://github.com/CalculatedContent/WeightWatcher
>
> It is fast, simple, and easy to integrate into any production MLOps pipeline.

---

### Decision · Program_Chairs · 2023-01-20

**Decision:**

Reject

**Justification For Why Not Higher Score:**

See metareview above.

**Justification For Why Not Lower Score:**

See metareview above.

**Metareview: Summary, Strengths And Weaknesses:**

This paper aims to compare and analyse neural models without having access to training or test data, offering new insights on a particular phenomenon, namely Simpson's paradox. The reviewers found the paper slightly difficult to follow for non-experts and the contribution incremental and somewhat niche. The authors contested the reviewers claims, however, reviewers were not convinced that the paper will have  any impact on wider research beyond the HT-SR theory.

**Summary Of Ac-Reviewer Meeting:**

This paper was not borderline.